# Crop HTP Technologies: Applications and Prospects

Shuyuan He [1,2], Xiuni Li [1,2], Menggen Chen [1,2], Xiangyao Xu [1,2], Fenda Tang [1,2], Tao Gong [1,2], Mei Xu [1,2], Wenyu Yang [1,2] and Weiguo Liu [1,2,*]

[1] College of Agronomy, Sichuan Agricultural University, Chengdu 611130, China; shuyuanhe479@163.com (S.H.)
[2] Key Laboratory of Crop Ecophysiology and Farming System in Southwest, Ministry of Agriculture and Rural Affairs, Chengdu 611100, China
[*] Correspondence: lwgsy@126.com

**Abstract:** In order to rapidly breed high-quality varieties, an increasing number of plant researchers have identified the functions of a large number of genes, but there is a serious lack of research on plants' phenotypic traits. This severely hampers the breeding process and exacerbates the dual challenges of scarce resources and resource development and utilization. Currently, research on crop phenotyping has gradually transitioned from traditional methods to HTP technologies, highlighting the high regard scientists have for these technologies. It is well known that different crops' phenotypic traits exhibit certain differences. Therefore, in rapidly acquiring phenotypic data and efficiently extracting key information from massive datasets is precisely where HTP technologies play a crucial role in agricultural development. The core content of this article, starting from the perspective of crop phenomics, summarizes the current research status of HTP technology, both domestically and internationally; the application of HTP technology in above-ground and underground parts of crops; and its integration with precision agriculture implementation and multi-omics research. Finally, the bottleneck and countermeasures of HTP technology in the current agricultural context are proposed in order to provide a new method for phenotype research. HTP technologies dynamically monitor plant growth conditions with multi-scale, comprehensive, and automated assessments. This enables a more effective exploration of the intrinsic "genotype-phenotype-environment" relationships, unveiling the mechanisms behind specific biological traits. In doing so, these technologies support the improvement and evolution of superior varieties.

**Keywords:** plant phenotype; digital plants; high throughput; deep learning





## 1. Introduction

Early measurements related to crops' phenotypic features were initially established using traditional methods, which, while relatively straightforward, were prone to significant errors. Moreover, these methods demanded significant human effort and time, often influenced by subjective factors. Given the rapid development of genomics, proteomics, metabolomics, bioinformatics, and big data computing, relying solely on traditional methods for studying plants' phenotypic traits is no longer sufficient for extracting data from the current vast datasets, which is detrimental to future agricultural development. In recent years, the emergence of HTP platforms has brought forth advantages such as efficiency, non-destructiveness, and full automation. Moreover, the phenotypic data obtained through high-throughput techniques encompass new features that are unattainable through traditional research methods. These features include convex hull area, convex hull vertices, compactness, grayscale values, top/side projection areas, top/side contour areas, and more. They offer additional possibilities for exploring the "genotype-phenotype-environment" relationships. HTP technology is expected to gradually replace traditional methods in the near future. Therefore, in order to achieve the efficient acquisition of crop phenotype data, it is particularly important to build a HTP technology platform.

This paper focuses on the research of crop HTP technology. We systematically introduce methods and applications related to HTP technologies, from the above-ground to the below-ground parts of crops, while discussing practical issues and solutions. We emphasize that the era of digital agriculture relies on analyzing crops through different systems to rapidly obtain multi-scale crop information, enabling researchers to have a more thorough understanding of crop growth conditions. This, in turn, facilitates the analysis of crop productivity from various angles. Digital agriculture plays a crucial role in the future development of agriculture, with HTP technologies serving as accelerators for breeding and precision agriculture [1], thus accelerating the selection of better varieties.

## 2. Overview of Plant HTP Research

In the context of "digital plants" technology, to gain a better understanding of the development of plant phenotypes, we collected articles on high-throughput techniques in plant phenotyping from Web of Science. As of 2023, there are over one hundred thousand research articles related to plant phenotyping, and the research in this field has shown a consistent upward trend since 1999 (Figure 1a). Among the various studies on plant phenotyping, a growing number of scholars prioritize HTP techniques. According to statistical data, HTP research has experienced rapid growth over the past two decades and has consistently maintained a high level of interest (Figure 1a), as evidenced by the number of studies presenting HTP research. Crop-related research leads the way in HTP research (Figure 1b). (The data in Figure 1 were sourced from the Web of science database, and the keywords for the first-level inspection were "Plant phenotype"/" High throughput plant phenotype ". The keywords for secondary retrieval were "crops"/" fruit "/" vegetable "/" forest "/" seed "/" phenomics "/" stems "/" leaves "/" flowers "/" roots "; the search date was set to the last 10 years.)

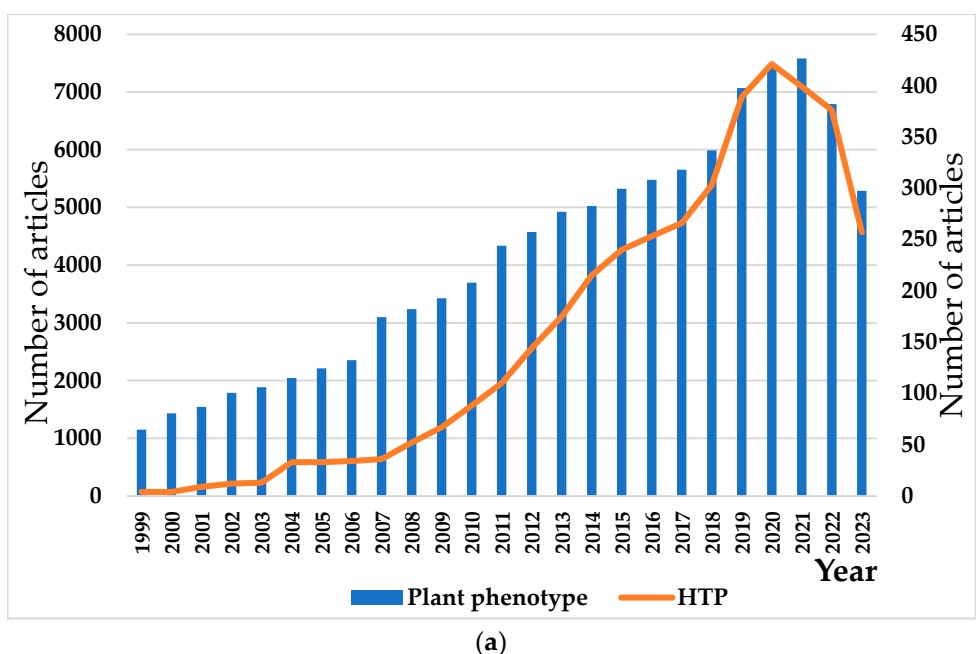

(**a**)

**Figure 1.** *Cont.*

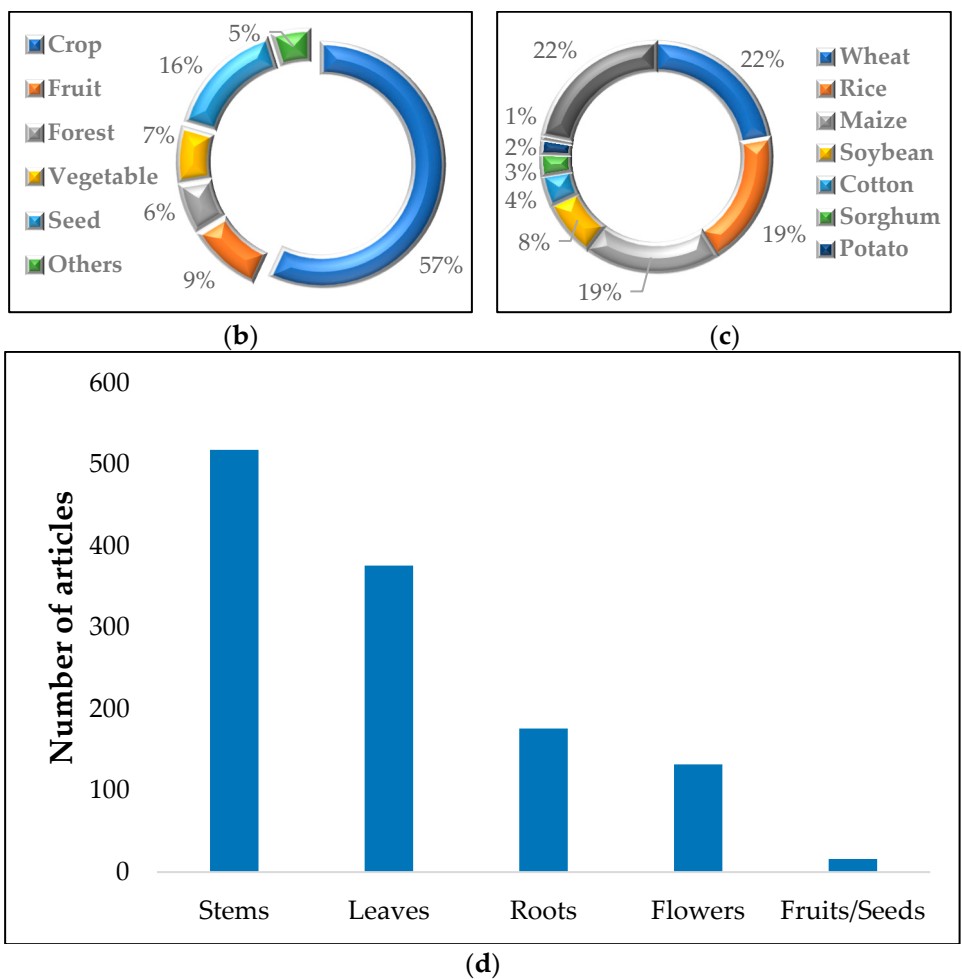

**Figure 1.** Studies on plant phenotypes in the past 20 years. (**a**) Statistics on the number of published articles on plant phenotypes and high-throughput plant phenotypes from 1999 to 2023. (**b**) Application of HTP techniques in plants (left). (**c**) Application of HTP technology in crops (right). (**d**) Application of HTP techniques to the above-ground and subsurface parts of plants.

We compiled data on the application of HTP techniques in major crops worldwide. The data reveal that crops such as wheat, rice, and maize are more widely studied in the field of HTP research. Following these are soybeans and cotton, while studies on crops like sorghum, potatoes, and peanuts account for less than 3%. We speculate that the primary reason for this phenomenon is that wheat, rice, maize, and sorghum are among the world's four major cereal crops, and the development of high-yield and high-quality varieties is crucial for addressing food security issues. Research on phenotypic traits accelerates the breeding process in these crops.

For sorghum phenotypic research, despite the tall stature of sorghum plants, existing HTP devices are capable of meeting the real-time monitoring needs of sorghum phenotypes throughout its entire growth period [2]. The plant architectures of dicotyledonous plants such as soybean and cotton are more compact and intricate compared to those of monocotyledonous plants. The setup of models and data accuracy for dicotyledonous plants requires constant refinement and adjustment, significantly delaying the application of HTP technology in dicotyledonous plants. In recent years, the application of HTP technology in dicotyledonous plants has seen a significant increase, driven by corresponding national policies and an increased emphasis on food security [3].

Furthermore, we also collected data on the application of HTP technology in both above-ground and below-ground plant parts since 1999. We found that the amount of research on above-ground parts significantly outweighs that on below-ground parts

(Figure 1d), which aligns with the findings of Solimani F. and others [4]. Specific practical applications and research advancements of HTP in both above-ground and below-ground plant parts will be discussed in detail in the fourth section of this paper.

According to a survey conducted jointly by the International Plant Phenotyping Network (IPPN), the European Plant Phenotyping Network (EPPN), and the German Plant Phenotyping Network (DPPN), several key challenges are currently impeding the development of plant phenomics. These challenges include field phenotyping, data management, costs, root phenotyping, abiotic stress, industry standards, technical limitations, bioinformatics, and throughput limitations. Among these challenges, field phenotyping is widely recognized as the most significant hurdle in plant phenotyping research and the development of plant phenomics. This is undoubtedly linked to the complexity and uncontrollability of field environments, which is why early HTP developments primarily took place indoors.

Nevertheless, we believe that the development and updating of field-based HTP platforms are essential because of their automation capabilities, enabling the real-time monitoring of dynamic changes in plant growth in field conditions. In early 2014, the International Crops Research Institute for the Semi-Arid Tropics (ICRISAT) officially launched the first truly commercial high-throughput plant field phenotyping platform, the FieldScan system, on an international scale. Currently, this platform is centered around the PlantEye plant laser 3D scanning instrument and integrates various other sensors. It autonomously collects plant phenotypic data around the clock in any environment, providing data of high precision. The introduction of this system marked a significant breakthrough in overcoming the challenge of obtaining field-based plant phenotypic data.

In general, there is increasing attention and emphasis on high-throughput plant phenotyping research both domestically and internationally, which holds promise for the future of agricultural development. Scientists are actively contributing to the rapid advancement of HTP technology by continually optimizing platform designs, integrating multiple sensors, efficiently extracting phenotypic features, enhancing data accuracy, and updating algorithm models. These research areas will be discussed in the next section.

## 3. Key Technologies for Obtaining HTP Information

High-throughput plant phenotyping platforms, which come in various forms, such as desktop, conveyor belt, autonomous, gantry, and drone-based platforms [3], are powerful tools in the fields of plant phenomics, plant functional genomics, and modern genetic breeding research. These platforms can be categorized into two main types based on their applicable environments: indoor platforms and field platforms. Additionally, they carry sensors with varying abilities to measure plant-related features, including RGB cameras, fluorescence cameras, laser radars, hyperspectral imagers, thermal infrared imagers, and more [5]. Previous studies have extensively applied HTP platforms to important crops such as rice [6,7], cotton [8], wheat [9], soybeans [10], rapeseed [11], sorghum [12], maize [13], and others.

Acquiring plant phenotypic information through HTP platforms involves three main steps: image acquisition, feature extraction, and data processing. Based on these three steps, we have compiled a significant body of literature and summarized it to gain a better understanding of how to efficiently, non-destructively, and automatically acquire extensive plant phenotypic data using HTP platforms.

### 3.1. Sensor Applications

The usage of sensors varies depending on the type of crop and features being measured. We categorize the acquired images into two-dimensional and three-dimensional images. By reviewing a substantial body of literature and summarizing information on common sensor applications, obtained features, and image properties, we were able to compile the details shown in Table 1. Currently, there is a relatively higher volume of research involving the setup of various phenotyping platforms using RGB cameras mounted on carriers such as

drones. These platforms capture crops from multiple angles, dimensions, and perspectives, allowing for the acquisition of images at different scales. With the rapid development of phenomics, many researchers are beginning to use multiple sensors in combination, when economically feasible, to acquire clearer and more reliable images.

**Table 1.** Phenotypic information obtained using common sensors and its applications.

| Imaging Technology | Phenotypic Information | Application | Scenario | References |
|---|---|---|---|---|
| Visible light imaging | RGB images | Plant height, canopy coverage, leaf to ear ratio | Field/Indoor | [14–17] |
| Fluorescence imaging | Sensitive bands | Drought stress, disease monitoring | Indoor | [18,19] |
| Three-dimensional imaging | Depth maps, point clouds, voxel data, grids, implicit data | Height, main stem length, leaf area | Field/Indoor | [13,20–22] |
| Infrared imaging | The continuous or discrete spectrum of each pixel | Stress monitoring, pest monitoring | Field/Indoor | [23–25] |
| Hyperspectral imaging | Continuous or discrete | Moisture monitoring, quality monitoring, pest monitoring | Field/Indoor | [26–28] |
| Multispectral imaging | Multiple bands of the spectrum | Chlorophyll, leaf area index, drought stress | Field/Indoor | [29–31] |

Visible light imaging is the most widely used technology in current HTP platforms. This is primarily because the sensors used are mainly low-cost RGB cameras that can be employed for both 2D image and 3D date capture and analysis, resulting in images that closely resemble human visual perception [32]. Currently, laser radars (LiDARs) serve as the primary sensors for acquiring 3D images. Compared to visible light cameras, LiDARs boast strong information-gathering capabilities, enabling the acquisition of more complete and intuitive 3D images with relatively high measurement accuracy and distance precision [33]. However, they may become inoperative in adverse weather conditions. Fluorescence imaging is primarily used indoors since it is easily influenced by the external light environment when gathering fluorescence parameter information. Infrared imaging primarily captures the heat radiated outward by the target plants. In comparison to visible light images, infrared images have lower resolutions, contrast, and signal-to-noise ratios, resulting in blurry visual effects and a non-linear relationship with target reflectance characteristics [34]. Nonetheless, infrared imaging excels in its ability to penetrate plant canopies and automatically monitor the growth dynamics of plants within the canopy under adverse conditions, with minimal sensitivity to lighting conditions [35].

In fact, both multispectral imaging and hyperspectral imaging are spectral imaging technologies. Hyperspectral imaging, due to its high spectral resolution, fast speed, lightweight, and low power consumption, can capture the brightness values of electromagnetic waves emitted or reflected by plants in various spectral bands, resulting in more refined and specific spectral data. However, hyperspectral imaging technology still faces many challenges during use, as the imaging process is significantly affected by external lighting conditions, leading to greater uncertainty in the obtained spectra [36]. On the other hand, multispectral imaging technology can quickly detect the spectral and radiative information of the target, providing a continuous spectral curve for each pixel within the imaging range using multiple channels. Therefore, it finds extensive applications in various fields [37]. However, having too many bands often leads to issues such as high redundancy in imaging spectral data, strong inter-band correlations, increased computational time, and large storage space requirements.

Research has shown that the complementary use of multiple sensors can increase data accuracy and reliability to a certain extent [38]. When integrated with wireless communication networks, it enables precise data collection within plant factories and the intelligent control of facilities, thereby significantly improving overall efficiency. With the continuous

advancement of image acquisition technology, we can expect to see more collaboration among sensors in future applications of high-throughput plant phenotyping platforms.

### 3.2. Phenotypic Information Extraction

After obtaining the images, the next step involves image processing and the extraction of features. This step is crucial in the process of obtaining plants' phenotypic information, and the quality of feature extraction directly determines the accuracy of the data. Image processing is the primary task of feature extraction. By using advanced image segmentation methods to remove complex backgrounds, scientists are able to preserve research objects and prepare for relevant feature extraction.

The subsequent task of image processing involves the extraction of specific phenotypic characteristics. The primary objective of feature extraction is to derive information with significant biological relevance, leading to the development of specific models tailored to the morphological structure and physiological and ecological functions of different plants. As phenomics advances, the methods for feature extraction will become increasingly diverse. Deep learning approaches stand out at this stage. Deep learning is a data-driven learning approach used to construct more intricate models, including Convolutional Neural Networks (CNNs), Recurrent Neural Networks (RNNs), Generative Adversarial Networks (GANs), and other model structures. CNNs have emerged as the predominant model type for most image recognition, classification, and detection tasks [39] (Table 2). (The data in Table 2 were sourced from the Web of science database. The keyword of the first-level retrieval was "High throughput plant phenotype". The keywords of secondary retrieval were "deep learning"/" convolutional neural network "/" recurrent neural network "/" generative adversarial network ".)

**Table 2.** Statistics of published articles on common deep learning algorithms.

| Model | Number of Study | Ratio (%) |
|---|---|---|
| Deep learning | 5967 | - |
| CNN | 3905 | 65.44 |
| RNN | 389 | 6.52 |
| GAN | 230 | 3.85 |
| Others | 1443 | 24.18 |

Note: CNN: Convolutional Neural Network; RNN: Recurrent Neural Network; GAN: Generative Adversarial Network; Others: other deep learning algorithms. Percentage of each project = Number of published articles for the project/Number of deep learning published articles $\times$ 100%.

The applications of deep learning method are widespread both domestically and internationally, and remarkable achievements have been made, mainly focusing on the extraction of three major features: morphology, color, and texture. For example, achievements have been made in the segmentation of crop canopy structures [40], extraction of leaf area [41], seed counting [42–44], and monitoring of plant growth conditions [45].

In recent years, 3D reconstruction technology has been widely applied in various fields, including gaming, filmmaking, clinical settings, autonomous driving, and virtual reality. Among them, research on image-based 3D reconstruction methods has been rapidly advancing due to their non-destructive nature, repeatability, low cost, practicality, and high accuracy, which are highly valued by researchers.

Currently, the field of 3D reconstruction uses five different types of picture data: implicit data, voxel data, grids, point clouds, and depth maps [46–48]. Each type has its unique reconstruction methods, and researchers utilize these methods to acquire the desired phenotypic information. In Figure 2, we present a summary of image-based 3D reconstruction methods categorized into two types based on the reconstruction principles: traditional multi-view geometry-based 3D reconstruction algorithms and deep learning-based 3D reconstruction algorithms. Traditional 3D reconstruction algorithms typically rely on low-level features in images, such as keypoints and lines. However, they may

face challenges in areas with single textures, unclear gradient changes, and complex and variable conditions, leading to issues such as a lack of adaptability and generalization.

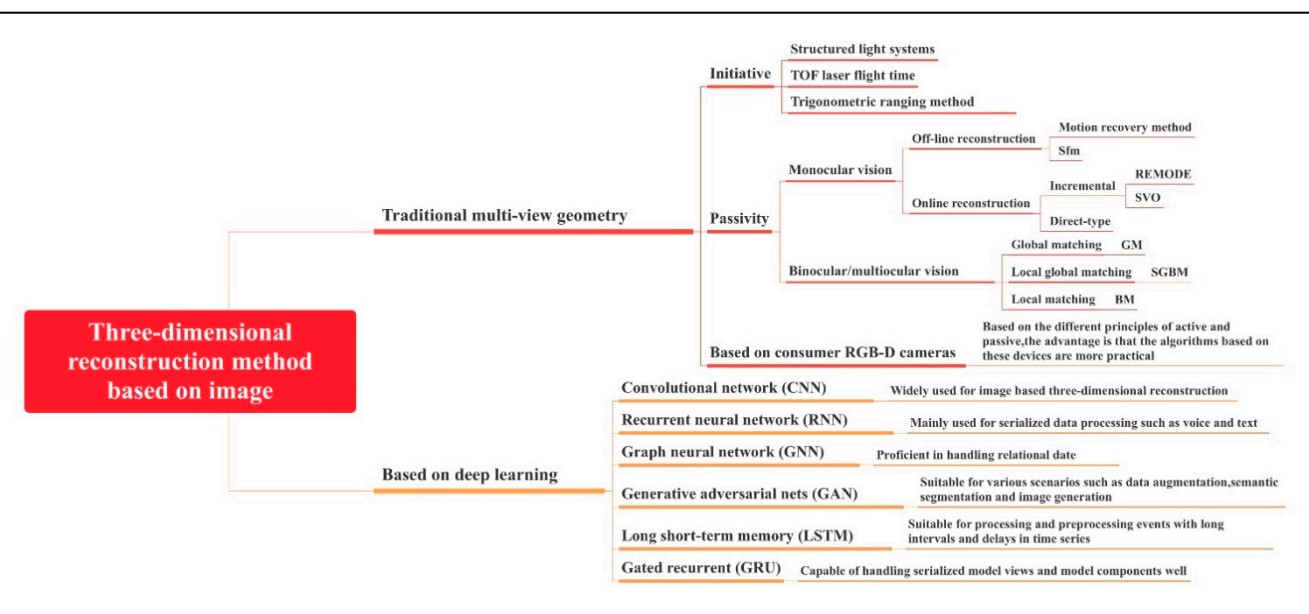

**Figure 2.** Image-based 3D reconstruction methods.

While traditional 3D reconstruction algorithms continue to play a dominant role in the field of 3D phenotypic research, an increasing number of scientists are transitioning from traditional methods to primarily adopting deep learning approaches. They are integrating deep learning reconstruction algorithms with traditional 3D reconstruction algorithms, leveraging the complementary strengths of both to provide new insights for optimizing the performance of 3D reconstruction algorithms. For instance, Xu et al. [49] captured multi-view images of individual wheat plants, resulting in a dataset of 1374 images. They used the Structure from Motion (SfM) method for 3D point cloud-based wheat plant reconstruction, employing a global calculation mode for one-time image processing to enhance efficiency while obtaining better 3D models. Furthermore, they conducted wheat plant point cloud surface reconstruction based on an improved Binary Partitioning Algorithm (BPA), modifying the method for identifying seed triangles and incorporating pertinent verification functions to enable the successful surface reconstruction of wheat point cloud models. Goodfellow et al. [50] introduced an Generative Adversarial Network structure consisting of a generator and a discriminator. Through adversarial training and optimization, the network achieved remarkable 3D reconstruction image results. The ReC-MVSNet algorithm proposed by Liu et al. [51] integrates phenotypic features into three-dimensional point cloud reconstruction grids, improving the accuracy by 43.3% compared to traditional MVS models.

Generally, for 3D reconstruction, deep learning methods such as CNNs, RNNs, and GNNs are commonly applied [52]. The choice of deep learning network frameworks can have a substantial impact on the development of 3D reconstruction, with different types being tailored to specific reconstruction tasks.

As HTP technology has advanced, images have remained the primary data format for plant phenotypic research. Feature extraction methods are predominantly rooted in deep learning [4]. Leveraging its robust feature extraction and modeling capabilities, deep learning has introduced innovative approaches for extracting crucial phenotypic information.

### 3.3. Phenotypic Big Data Processing

In contemporary society, phenotypic data are experiencing an astonishing surge in interest. Unlike traditional data, HTP data are characterized by their diverse and complex natures. These data are often unordered, randomly sized, and analyzed using varying methods. In contrast to traditional data processing methods, the analysis of HTP data imposes even more stringent requirements on data access, processing, and analysis. Consequently, efficiently handling massive data has become one of the crucial tasks in the development of HTP technology.

In the face of vast amounts of phenotype data, big data technology emerges as a comprehensive solution for data storage and processing. It is well suited to handling and analyzing high-capacity data, facilitating efficient data storage management, and enabling data sharing across different systems. Common big data processing technologies include MapReduce, Hadoop, and Spark. With the widespread adoption of HTP technology, combined with the complexity of the external environment and the genetic diversity of species, a significant amount of data exhibit distinct characteristics known as the 3Vs (i.e., volume, variety, and velocity) and the 3Hs (i.e., high dimensionality, high complexity, and high uncertainty) [3].

Priyadharshini et al. [53] utilized the LeNet network to classify the severity of corn leaf diseases, achieving a high model accuracy of 97.89%. Mehedi Hasan et al. [54] employed a region-based Convolutional Neural Network (R-CNN) to accurately identify and count wheat spikes, subsequently using the same network architecture to build four models, all of which effectively predicted spike yield with an average detection accuracy ranging from 88% to 94%. Li et al. used a Unet neural network to segment soybean images after performing RGB imaging on 208 soybean materials. The segmentation impact was good, and the values of IOU, PA, and Recall could approach 98%, 99%, and 98%, respectively [55].

At present, crop phenotyping big data technologies and equipment are undergoing rapid development. This development is fueled by the demands of agricultural research and production and is instrumental for advancing the crop phenotyping industry [56]. The ongoing evolution of the phenotype feature extraction process has significantly propelled the advancement of high-throughput plant phenotyping research. This evolution fundamentally addresses challenges associated with traditional techniques, such as challenges pertaining to labor intensiveness and time consumption. Moreover, it enables the rapid extraction of valuable information from massive phenotype data, holding immeasurable potential for the future development of agriculture.

## 4. Application of Crop HTP Technology

In recent years, HTP technologies have primarily focused on various morphological, textural, and color features of plants, with relatively less attention being paid to physiological and biochemical indicators. Currently, the determination of plants' physiological and biochemical indicators still relies mainly on traditional methods. Scientists from various fields should pay more attention to the study of physiological and biochemical mechanisms and integrate plant morphology with physiological and biochemical characteristics. This interdisciplinary approach will accelerate the breeding process. Research on the monitoring of the growth and development of above-ground and below-ground parts of crops through HTP technologies has gained increasing attention. Detailed discussions on these topics are provided in Sections 4.1.1 and 4.1.2 of this paper.

### 4.1. Monitoring the Growth and Development of Crops

#### 4.1.1. Crops' Above-Ground Portion

The above-ground portion of plants produces assimilates through photosynthesis for the plant's absorption and utilization, forming the material basis for plant growth and development. The quality of the above-ground plant canopy structure not only determines the efficiency of solar energy utilization but also affects yield stability and cultivation techniques. As a result, articles on crop canopy structures have been increasingly emphasized,

whether in the study of individual plant morphology or population structure. In HTP research on crops, features such as plant height, canopy structure, leaf area, and vegetation indices have garnered significant attention from numerous researchers. Therefore, this article primarily elaborates on the research progress regarding the use of HTP technology for determining these four features.

In the past, the acquisition of stem phenotype information in plants primarily focused on features such as plant height, stem diameter, internode number, and internode length. Research has shown that plant height is a key quantitative descriptor of dynamic growth and development differences among crop varieties. It is also considered a core phenotypic feature in the field of crop breeding, with a significant relationship to the construction of ideal plant architectures and crop yield components [57].

Sun et al., used a tractor-mounted laser scanner to scan cotton plant heights, achieving an R-squared value of 0.98 [21]. The same team [22] employed high-resolution 3D ground-based laser scanning to detect cotton main stems and nodes. This method accurately measured cotton main stem length with an R-squared value of 0.94. In order to determine the phenotypic attributes of a single 3D maize plant, Ao et al. [58] combined a laser radar with the PointCNN model to obtain 3D phenotypic traits related to individual maize plants, achieving an $R^2 > 0.99$ between the measured and actual values of plant height.

In summary, the use of high-throughput 3D phenotyping technology for measuring plant height has yielded high coefficients of determination. This suggests that 3D phenotyping technology enables the acquisition of more accurate phenotypic information, and its establishment will play a crucial role in the future advancement of HTP technologies.

The crown structure of plants essentially encompasses the spatial arrangement of the above-ground part of plants. Research has demonstrated that the crown structure better reflects the true structural characteristics of crops [59] and is significantly positively correlated with crop yield. In recent years, scientists have concentrated on the study of crops' crown structure using different sensors and models, primarily focusing on two features: crown height [60] and crown coverage [61].

Casagrande et al. [16] used a drone-mounted RGB camera to capture field images and calculated crown coverage for soybeans during the V3-R1 stage based on the ratio of green pixels for each experimental unit. The results showed that, as the soybean growth stage progressed, soybean yield was positively correlated with canopy photosynthesis, with a correlation of 0.76 at the V9-R1 stage. Borra-Serrano et al. [14], addressed the challenge of filling high-resolution objective data gaps in the growth stages of soybeans in a time-series manner, achieving an accuracy of over 90% for soybean canopy height compared to traditional methods.

Currently, the exploration of features related to canopy structure is ongoing. While there are fewer features associated with canopy structure compared to individual plant structures, their importance remains significant. There is a growing belief among scientists that population structure better reflects the inherent growth dynamics of plants, emphasizing the need for research on canopy structure. This research plays a vital role in the pursuit of ideal plant architectures across various crop types.

Traditionally, vegetation indices have been crucial indicators for measuring population coverage. Conventional canopy analysis methods rely on manual, handheld measurements taken at the observation site. While this traditional approach is suitable for small-scale and infrequent measurements, it becomes insufficient when larger spatial areas and higher temporal frequencies of monitoring are required.

The vegetation index measures the state of vegetation development under specific circumstances by comparing the reflection of vegetation in visible and near-infrared bands with the soil background. The variation in vegetation indices is controlled by the interaction between its own genotype and the environment [62]. Alexander J. Lindsey et al. [63] used the Normalized Difference Vegetation Index (NDVI) to assess soybean canopy senescence and determined the observed maturity of each variety, thus calculating the maturity period of soybeans.

Currently, both domestic and international research is increasingly utilizing HTP technologies for the real-time monitoring of crop vegetation indices. For instance, Guo et al. [64] used a 3D laser sensor (Planteye F500) to identify and monitor normalized vegetation indices at multiple time points in maize under normal and saline conditions, accurately quantifying the morphological traits of maize seedlings in different growth environments. Christopher et al. [65] developed a vegetation spectrometer (TSWIFT) capable of continuously and automatically monitoring hyperspectral reflectance, enabling the assessment of changes in soybean structure and function at high spatiotemporal resolutions, thus achieving HTP values.

The Leaf Area Index (LAI) is a crucial quantitative feature for describing vegetation, being closely related to photosynthesis, respiration, and transpiration processes. In recent decades, high-throughput techniques have been used to measure the LAI of crops such as maize [66], cotton [67], and rapeseed [68]. Zheng et al. [69] developed a new unified linearized vector radiative transfer model (UNL-VRTM) based on multispectral data. The forward modeling of the model has a strong coupling between vegetation canopy and the atmospheric environment, and the simulation process is reasonable, which means it can support the synchronous detection of the LAI and Cab. In 2021, Feng Xiao and colleagues introduced a new feature called the "leaf panicle ratio" to describe the new light interception characteristics of hybrid rice based on the Leaf Area Index [17].

4.1.2. Crops' Underground Components

The underground part of plants, i.e., the root system, absorbs mineral elements and water from the soil to meet the needs of above-ground processes such as transpiration and various physiological functions. A robust and well-developed root system is the foundation for crop growth and high yields, and shaping the root architecture of a population contributes to increased yields. Most plant roots are deeply embedded in the soil, making it challenging to observe the dynamic changes in root growth in real time. Traditional root phenotype studies often rely on monitoring above-ground growth conditions to indirectly reflect the growth of underground parts (roots), leading to significant errors and hindering the development of root phenotype research.

Moreover, previous root studies often involved destructive sampling, which has a detrimental impact on the subsequent growth and development of plants, ultimately compromising yield. Therefore, the primary challenge in root phenotype development is how to collect root phenotype information non-destructively. This challenge has underscored the emergence of HTP technologies. In 2016, Jeudy et al. [70] proposed the use of RhizoTube, a root tube for plant cultivation. Seeds are placed between an external transparent tube and a physiological membrane, allowing for root imaging while cultivating plants. This method enables the non-destructive collection of root phenotype information. Two years later, Delory et al. [71] used a root box to measure plant root phenotypic traits, enabling the non-destructive, real-time monitoring of roots under transparent conditions. However, the limited size of root tubes and root boxes makes them unsuitable for larger plant roots. In reality, the roots of crops grow in an opaque environment, and the interaction between roots and the soil matrix is strong. Therefore, how to non-destructively visualize and quantify root growth in opaque soil has become a difficult problem in root phenotype research.

Techniques for root imaging currently include visible light imaging, MRI, and X-ray CT. Visible light cameras are often preferred due to their low cost and convenience, but they typically yield poor root images and provide limited root phenotype information. As a result, they are frequently used in combination with other sensors [72]. MRI and X-ray CT technologies allow for the three-dimensional, non-destructive observation of underground root structures, offering a more intuitive and comprehensive approach to root phenotype research. However, not all soil matrices are suitable for MRI, as their suitability depends on factors such as size, moisture content, nutrients, root thickness, and root area in the soil [73]. Therefore, MRI technology has extensive requirements. On the other hand, X-ray CT technology uses a precise collimated X-ray beam, gamma rays, ultrasonic waves, and

highly sensitive detectors to scan sections around a certain part of the test object, providing fast scanning times and clear images. It is commonly used in the medical field. When studying the specific details of fine roots within a certain planting area diameter, X-ray CT has a higher spatial resolution than MRI. However, when the area exceeds a certain size, MRI provides greater root quantities than CT technology [74].

In summary, enhancing and breeding varieties with desirable root traits offers strong theoretical support for increasing crop yields. However, compared to above-ground phenotype studies, the application of high-throughput technology to the underground part is less common. This is mainly due to the challenges of obtaining materials, imaging difficulties, and high costs. The study of the underground part presents significant challenges which current HTP technologies cannot fully address, often requiring technologies such as CT technology, MRI, and ultrasound.

## 4.2. Crop Yield Prediction

In the 21st century, research on plant fruits, particularly crop seeds, faces two main challenges: increasing yield and enhancing seed quality. It is evident that simultaneously improving both aspects is quite challenging, making exploring coordinated enhancements a key issue for researchers. This requires the consideration of various aspects of plant growth habits, the environment, and usage and the integration of multidisciplinary expertise to explore the best practices for synergistically improving yield and quality.

Food security and other related concerns are benefited by accurate crop production forecasting, which also offers significant insights into agricultural progress [75]. This is why agricultural production forecasting research is highly regarded both domestically and internationally. The primary characteristics utilized to determine yield in previous research on seed phenotype were seed appearance and weight, with a focus on the harvest time following maturity. Nevertheless, it was not possible to anticipate yield prior to maturity, which posed significant problems to crop yield prediction and did not help to improve output.

In today's era of high-throughput technology, many researchers are developing various system models and methods for the identification and prediction of various plant fruits (seeds) before maturation, as illustrated in Table 3. The process of fruit/seed identification and prediction mainly involves the following steps: obtaining seed images, image processing and feature extraction, importing corresponding analysis tools, measuring using different models for different tasks, training and evaluating the selected models, analyzing data, and obtaining seeds' phenotypic traits [76]. Currently, the high-throughput acquisition of fruit/seed phenotypic information is concentrated on researching seed number, purity, vitality, etc. Regarding the determination of seed maturity, researchers have also begun to conduct studies in this area [77].

**Table 3.** Suitable yield monitoring models for different crops.

| System Model | Sensor | Algorithm | Application Area | Accuracy (%) | $R^2$ | References |
|---|---|---|---|---|---|---|
| YOLO-v4 | RGB | CNN | Wheat | 96.04% | - | [78] |
| YOLO-v5 | RGB | SM-YOLOv5 | Tomato | 97.8% | - | [79] |
| YOLO-v8 | NIR, UV, RGB | YOLO-v8n | Tomato | 65.08% | - | [80] |
| YOLO X | - | - | Technological update | Increased to 45.0% AP | - | [81] |
| YOLO POD | RGB | CBAM | Soybean | - | 0.967 | [82] |
| X-rayCT | X ray CT scanning system | MATLAB | Rice | - | 0.98 | [83] |
| P2PNet-Soy | RGB | Unsupervised clustering | Soybean | - | 0.87 | [84] |
| TasselGAN | RGB | DC-GAN variant | Maize | 72.36% | - | [85] |
| Anchor-free ObjectBox | RGB | CBAM | Wheat | 94.5% | - | [86] |

### 4.3. Crop Growth Environment Monitoring

Previous studies have indicated that a favorable planting environment contributes to increased yields, showcasing the effectiveness of cultivation practices. In the plant kingdom, stress mainly manifests in two forms: biotic stress and abiotic stress. Both types pose varying degrees of threats to the growth and development of plants, representing major factors hindering the efficient development of agriculture. As we all know, all biological elements that are detrimental to plant growth and development, such as diseases, pests, and grasses, are collectively referred to as biological stress. Any abiotic condition that negatively impacts plant growth, such as drought, flooding, salinity, nutrient shortage, and so forth, is referred to as an abiotic stress.

In the past, the diagnosis of crop symptoms often relied on human observation, a method that, due to its inability to facilitate prompt observations, led to the optimal treatment window being missed. This limitation constrained the precision of crop stress management, ultimately impacting the growth and yield formation of crops.

In the current stage, the real-time monitoring of dynamic changes in plant growth using HTP technology allows for the identification of the optimal periods for stress management. This facilitates stress management measures and precise resource allocation, providing robust support for the high-throughput screening and identification of superior stress-resistant varieties. Up to now, researchers have developed specific monitoring models that can be applied to different crops, primarily focusing on rice [87], wheat [88], and others [65,89]. Forecasting crop weather conditions in advance is beneficial for the implementation of crop cultivation management and biological control. Meteorological data collected through big data are first sorted into high-resolution data information through Earth System Models (ESMs). Then, this information, combined with machine learning tools, provides forecasts of weather conditions for the upcoming 15 days in the respective regions, as well as warnings for impending extreme weather events [90]. This is crucial for monitoring the crop growth environment.

Researchers define canopy features through image features and plant traits and then use such features to rationally adjust conditions related to water, nutrients, air, and heat. The team led by Thorp found that multispectral images can be used to determine crop canopy coverage and estimate crop coefficients, thereby improving water use efficiency [91]. Naik et al. [92] utilized unmanned aerial vehicles equipped with multispectral infrared thermal imagers to capture soybean field images. They extracted five image features, including canopy temperature, the Normalized Difference Vegetation Index, canopy area, canopy width, and canopy length, effectively assessing the extent of waterlogging disasters in the population.

The application of high-throughput technologies holds an indispensable position in precision agriculture, particularly in the crucial aspect of monitoring the plant growth environment. Therefore, in the era of digital plants, it is not surprising that high-throughput platforms are being used for the real-time monitoring of the microenvironment of plant growth.

### 4.4. Integration of HTP and Multi-Omics in Crops

Over the years, the inherent genetic diversity of crops has constrained the progress of crop breeding, thereby delaying the development of new crop varieties. Relevant research indicates that crop-improving omics technologies include genomics, transcriptomics, proteomics, metabolomics, and phenomics, among others [93]. Studies combining crop phenomics with multi-omics approaches have been well established in crop science. By utilizing machine learning (ML) and deep learning (DL) techniques to construct predictive models, the efficiency of crop improvement approaches such as genomic selection and genome editing has been enhanced, playing a significant role in crop growth, yield, and responses to abiotic stresses.

A plant phenotype (P) is the result of the genotype (G), environment (E), and their interaction (GxE). Prediction in genomic selection models (GS) is based on the mathematical relationship between genotype and phenotype data from the target population. Typically,

the relationship between a single phenotypic trait and genotype can be modeled using regression models [94].

In order to select phenotypes with outstanding target traits from among multiple traits, various genetic models have been created to accurately predict phenotype traits. Among them, the most typically used ones are Best Linear Unbiased Prediction (GBLUP) and Bayesian (BN) models. Montesinos-Lopez et al. [95] found that when considering genotype–environment interaction terms, the GBLUP method has the best genomic prediction performance compared to multi-trait deep learning models (MTDLs). Cantelmo et al. [96] used Dart-seq markers associated with the additive dominant genomic Best Linear Unbiased prediction (GBLUP) model for genome-wide selection and obtained a correlation of more than 0.82 between the predicted value and the actual value. Matei et al. [97] used a Bayesian model to predict the genotypes related to soybean yield traits, with mean allele frequency (MAF) values of 0.6591, 0.9877, and 0.0205 for the average, maximum, and minimum, respectively, indicating high accuracy.

As is well known, the development of an excellent variety often relies on the cross-fusion of multiple omics, thus accelerating the improvement of desirable agronomic traits in crops. In recent years, the emergence of high-throughput multi-omics technologies has fundamentally transformed crop breeding research, often containing rich data resources. Researchers have now organized and analyzed multi-omics data through different databases. For instance, Gui et al. [98] constructed the comprehensive ZEAMAP database for maize multi-omics research; Yang et al. [99] built a multi-omics database for rapeseed that includes genomics, transcriptomics, mutomics, epigenomics, phenomics, and metabolomics datasets; Gong et al. [100] developed the GpemD8 database, which has been successfully applied in studying excellent traits in rice populations. The integration of omics data will provide a comprehensive explanation of the interactions between crop traits, marking a crucial step towards enhancing the breeding of excellent varieties and addressing food security issues [101].

In today's era of multi-omics development, we advocate for the integration of crop phenotypes with multi-omics, starting from "phenotype-genotype" and ultimately returning to "genotype-phenotype" research. In this cyclical process, relevant predictive models are established to accelerate the improvement of crop breeding.

## 5. Challenges and Strategies for Crop HTP Technology

HTP technology has achieved significant success in the field of plant phenotyping research, especially in crop phenotyping studies. The combination of plant phenotyping and breeding has promoted the rapid development of digital agriculture, increasing people's attention to the development of this innovative approach. In recent years, scientists have conducted HTP analysis on subjects ranging from individual plants to populations, further refining it to various organs, tissues, and cells, laying a solid theoretical foundation for improving crop traits and increasing yields.

### 5.1. Simple of Phenotypic Information

The construction of ideal plant architectures relies heavily on extensive data support [102]. Using traditional methods to build ideal plant architectures presents unexpected challenges, and a continued reliance on traditional approaches is not conducive to the future of agriculture. So far, researchers in various crop research fields have employed different system models and methods to non-destructively analyze crops from multiple angles and perspectives, rapidly acquiring and analyzing crop phenotypic features [103], laying a solid foundation for further modifying plant architecture and constructing ideal plant architectures. Moreover, considering the bottlenecks regarding the future of grain production, continuous improvements and modifications towards ideal plant architectures will be a key focus of agricultural development strategies.

The development of crop phenomics has faced the challenge of information singularity, which manifests in two main aspects: the limited use of sensors on one hand [5] and the

uniformity of model construction, particularly in the field of root phenotyping, on the other hand. Both of these factors contribute to the homogenization of phenotypic data acquisition.

As previously mentioned, unmanned aerial vehicles (UAVs) equipped with sensors have been widely used in plant phenotyping research. However, these UAVs have limitations in terms of the number of sensors they can simultaneously carry. Most commonly, they employ single sensors such as RGB cameras, infrared imaging, multispectral or hyperspectral cameras, and other sensors for remote sensing analyses of crop phenotypes, maturity assessments, pest and disease diagnoses, growth monitoring, yield estimations, and analyses of key phenotypic features. The use of sensors in this context is still in its early stages. Many of these sensors tend to be expensive, which has restricted the development of high-throughput plant phenotyping research at this stage. Scientists are actively seeking low-cost, high-resolution sensors, but in reality, sensors that possess both of these characteristics are challenging to find [3]. Research has shown that using multiple sensors for plant phenotyping can improve accuracy compared to using a single RGB camera alone [38]. Therefore, in the context of the digital plant era, the emphasis is on establishing phenotyping platforms with multiple sensors.

In the current trend of agricultural development, in order to let artificial intelligence algorithms solve practical problems or to develop in the direction of strong artificial intelligence, simple model algorithms are certainly needed, and almost everywhere can be seen to break the dimensional wall for integration, so the direction that still has the potential for breakthroughs pertains to the super-large complex network structure; secondly, there are certain differences in the construction of various crop models, and the lack of a unified model structure suitable for various plants has led to a lack of communication among researchers, which is not conducive to the development of diversified data information. We call for the construction of multi-sensor phenotypic platforms to diversify phenotypic information through the continuous optimization of system models.

### 5.2. Instability of Data Accuracy

The use of high-throughput platforms in the study of plant architecture has reached maturity, but the significant research task of improving data accuracy continues. Phenotypic feature extraction is influenced by multiple factors, leading to fluctuations in data accuracy within a certain range. Therefore, we should gradually enhance data accuracy and stability by considering the growth environment of crops. The stability of data accuracy in field environments is lower compared to indoor settings, primarily because the growth environment of plants in the field is complex and variable. As it is well known, the more significant the changes in external environmental conditions and the more complex the plant architecture, the more pronounced the characteristics of the "3Vs" (i.e., volume, variety, and velocity) and "3Hs" (i.e., high dimensionality, high complexity, and high uncertainty) become, which also applies to the stability of data accuracy for other phenotypic extractions. Therefore, researchers should focus not only on improving data accuracy but also on ensuring data stability. Improvements in data accuracy and stability are conducive to the steady advancement of HTP technology.

Regarding enhancing data accuracy, we propose the following recommendations: Firstly, in the event of a controlled crop growing environment (e.g., indoor growth), supply the best growing medium for the crops, guarantee uniformity in every processing environment, and reduce errors resulting from environmental factors; second, the idea is to maintain the typical growth and development of crops, set up numerous repeated experiments, and gradually increase the prediction accuracy by continuously updating and optimizing the model when the growth environment of crops is artificially uncontrollable (such as field growth). Researchers have developed mixed phenotypic multi-trait (binary, ordered, and continuous) models, which have achieved moderate prediction accuracy improvements compared with univariate models (UDLs) and mixed phenotypic multi-trait models (MTDLMPs) in continuous traits [104].

*5.3. Non-Uniformity of Data Formats*

Plant databases serve as core repositories for various plant data, containing a wealth of information. However, there is significant variation in the contents of these databases. Based on their differences in stored data, databases can be categorized into three types: plant phenotype databases, distributed phenotype data collection, and information management systems for environmental data (Crop Sight) [105]. In terms of plant phenotype data, the predominant system is the Multi-Source Multi-Scale Phenotype Information Hybrid System (PHLS) [106], considering a diverse array of phenotype data formats, which hinders the smooth storage and analysis of data for researchers. With the advancement of breeding work, massive amounts of phenotype data have emerged, although not all of these data are useful. The complexity of data information makes it challenging for individuals to process such data immediately, potentially leading to significant data resource waste, which is not desirable in the scientific research process. Currently, there is a growing awareness of the essential need to efficiently extract and process relevant data information in phenotype research.

To overcome the aforementioned challenges and aid the future development of phenomics, there is an urgent need to establish a dedicated plant phenotype database. This would serve as the foundation for a shared platform for phenotype resources, allowing for the standardization of data information. This approach is beneficial not only for preserving data but also for facilitating mutual learning and research among scholars. Additionally, effective communication and collaboration among interdisciplinary researchers, as well as sharing research outcomes, are prerequisites for establishing a unified framework for plant models [107].

The integration of multi-omics and interdisciplinary collaboration is an essential path for the development of various disciplines, a crucial avenue for fostering innovative talent, and a rapid channel for breeding superior varieties. This is also indispensable for the advancement of HTP technology. We call for interdisciplinary integration, encourage the cultivation of interdisciplinary talents, and support the flourishing development of HTP technology in the context of digital plants.

## 6. Outlook

With the development of genetic, gene expression product, and various genetic association analysis studies, plant phenotypes have gradually attracted people's attention and evolved into an indispensable research area in biology. In the past decade, plant phenotypes have been clearly defined: they are considered to be physical, physiological, and biochemical mechanisms that reflect the structural and functional characteristics of plant cells, tissues, organs, plants, and populations. Essentially, phenotypes represent the three-dimensional expression of a plant's genetic map, with regional differentiation features and evolutionary processes [108–110].

It is projected that the global population will reach 11 billion in the next 25 years, leading to a substantial increase in food demand. In order to ensure food security, it is essential to utilize new technologies and methods such as HTP techniques to breed high-yielding and high-quality crop varieties.

Currently, the development of this technology is more mature in foreign countries, mainly in France, Germany, the Netherlands, Australia, and other countries. Comparatively, domestic phenotypic research development is lagging behind [1], but this does not mean that we should not attach importance to it. It is well known that innovation in methods and the construction of technology are the core challenges to applying HTP technology to crops. Relying solely on foreign imports is not conducive to the development of crop phenotyping technology. Instead, we should focus on technological innovation; adhere to the principle of emphasizing both hardware facilities and software technology; strengthen academic exchanges at home and abroad; prioritize multidisciplinary talent cultivation; and advocate for diversified phenotypic information, stable data accuracy, and standardized data sharing.

We firmly believe that in the future of agricultural development, HTP technology will gradually replace traditional methods and take a leading role. By continuously optimizing system models, enhancing data accuracy, and efficiently utilizing data information, we can accelerate the acquisition of plant phenotypic information, thereby contributing to the development of breeding and agriculture. With the development of phenomics, we have successfully overcome the challenge of transitioning from machine learning to deep learning. Deep learning is utilized extensively and is the subject of many different research areas. It is essential to the study of plant phenotypic data because of its substantial advantages in feature extraction and data processing [111]. Building on this foundation, efforts should be intensified to explore the application of deep learning technology in high-throughput phenotypic research, thereby addressing the technical bottlenecks we currently face.

**Author Contributions:** Writing—original draft preparation and software, S.H.; data curation, S.H. and M.C.; visualization, S.H. and F.T.; supervision, M.X. and W.Y.; formal analysis, T.G.; project administration, X.L.; writing—review and editing, S.H. and X.X.; financial support and overall review, W.L. All authors have read and agreed to the published version of the manuscript.

**Funding:** This work was supported by the National Natural Science Foundation of China (32172122) and the National modern agricultural industrial technology system of Sichuan Department of Agriculture and Rural Affairs (SC-CXTD-2020-20).

**Acknowledgments:** The authors thank the anonymous reviewers and journal Editor for their valuable suggestions, which helped to improve the manuscript. And we would like to thank Teacher Xu Mei for helping with the work and Teacher Liu Weiguo for helping to revise and improve the article.

**Conflicts of Interest:** The authors declare no conflicts of interest.

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
