# Peer review of "Crop HTP Technologies: Applications and Prospects"

_agriculture, doi:10.3390/agriculture14050723_

Round 1

Reviewer 1 Report (Previous Reviewer 2)

Comments and Suggestions for Authors

The authors have significantly improved the manuscript since my initial review. To demonstrate a greater mastery of the subject, I suggest modifying the term 'HTP' throughout the manuscript. Instead of using 'high throughput phenotype,' please use 'high-throughput phenotyping' from the title and consistently throughout the text (ex., Fig 1a, and description).

Author Response

Thank you for your valuable suggestions amidst your busy schedule. We have made modifications to the relevant content of the article as per your requirements. The specific revisions have been highlighted in red font in the revised manuscript. We hope these changes will enhance the quality of our manuscript.

We will now summarize the detailed modification instructions in the revised manuscript (i.e. corresponding responses to each modification suggestion), as shown below:

Comments 1: The authors have significantly improved the manuscript since my initial review. To demonstrate a greater mastery of the subject, I suggest modifying the term 'HTP' throughout the manuscript. Instead of using 'high throughput phenotype,' please use 'high-throughput phenotyping' from the title and consistently throughout the text (ex., Fig 1a, and description).

Response 1: Thank you for the suggestions from the expert reviewers. The issues you raised regarding our manuscript have been immensely helpful in improving its quality. We have completed the modification of all instances of the term "HTP" throughout the entire manuscript (including the language used in figures and their related descriptions), providing logical coherence to the structure of the article and ensuring content consistency.

Once again, we would like to express our gratitude to all the expert reviewers. Your support is highly valuable to us.

We look forward to hearing from you again.

Wishing you smooth work and a happy life!

Reviewer 2 Report (New Reviewer)

Comments and Suggestions for Authors

The article: Crop high throughput phenotype: Technologies, Applications and Prospects, discusses fomentation methods from traditional to high throughput technologies (HTP) in agriculture. It highlights the efficiency and automation of HTP, which are vital to modern agricultural development. The article highlights the importance of integrating HTP with precision agriculture. The authors suggest that HTP technologies can significantly accelerate breeding and improve crop varieties. The article discusses applying HTP technologies for above- and below-ground plant parts. The article provides an overview of the current status and future directions of HTP technology in agriculture, making it a valuable contribution to the field. However, it would be worthwhile to expand the article with examples of the practical application of HTP in various plant crops. In addition, expanding the discussion to include economic aspects of large-scale HTP implementation could provide a more holistic picture of its potential use in agriculture. The article is well-written and can be used in developing agricultural research. The article mentions the challenges of data accuracy and stability, indicating that the technology may still have limitations for use under field conditions. The article should be expanded with a conclusion on the practical use of high-quality phenotyping methods and phenotyping traits of plants. Errors in the text should be corrected before publication.
The language of the article 'Crop high-throughput phenotype: technologies, applications and perspectives' is technical and specialised. It is aimed at an audience familiar with agricultural science, particularly crop phenotyping. Despite the specialised language, the article maintains clarity in explaining the importance of high-throughput phenotyping technologies in agriculture. The article effectively communicates the complex concepts of crop phenotyping and its implications for agricultural development.

Author Response

Thank you for your valuable suggestions amidst your busy schedule. We have made modifications to the relevant content of the article as per your requirements. The specific revisions have been highlighted in red font in the revised manuscript. We hope these changes will enhance the quality of our manuscript.

We will now summarize the detailed modification instructions in the revised manuscript (i.e. corresponding responses to each modification suggestion), as shown below:

Comments 1: The article: Crop high throughput phenotype: Technologies, Applications and Prospects, discusses fomentation methods from traditional to high throughput technologies (HTP) in agriculture. It highlights the efficiency and automation of HTP, which are vital to modern agricultural development. The article highlights the importance of integrating HTP with precision agriculture. The authors suggest that HTP technologies can significantly accelerate breeding and improve crop varieties. The article discusses applying HTP technologies for above- and below-ground plant parts. The article provides an overview of the current status and future directions of HTP technology in agriculture, making it a valuable contribution to the field. However, it would be worthwhile to expand the article with examples of the practical application of HTP in various plant crops. In addition, expanding the discussion to include economic aspects of large-scale HTP implementation could provide a more holistic picture of its potential use in agriculture. The article is well-written and can be used in developing agricultural research. The article mentions the challenges of data accuracy and stability, indicating that the technology may still have limitations for use under field conditions. The article should be expanded with a conclusion on the practical use of high-quality phenotyping methods and phenotyping traits of plants. Errors in the text should be corrected before publication.
The language of the article 'Crop high-throughput phenotype: technologies, applications and perspectives' is technical and specialised. It is aimed at an audience familiar with agricultural science, particularly crop phenotyping. Despite the specialised language, the article maintains clarity in explaining the importance of high-throughput phenotyping technologies in agriculture. The article effectively communicates the complex concepts of crop phenotyping and its implications for agricultural development.

Response 1: Thank you for the high appreciation of our work by the expert reviewers. Your recognition is highly important to us, and we hope this article can provide a research direction for phenotypic researchers. The future development of agriculture will accelerate the study of plant phenotypic traits through continuous optimization of systemic models, improving data accuracy, and effectively utilizing data.

After careful consideration, we believe that the interdisciplinary integration of multi-omics and multiple disciplines has facilitated academic development, which is indispensable for research in various fields, especially in the realm of high-throughput phenotyping of crops. Therefore, in the concluding section, we once again emphasize the content of interdisciplinary integration and development (see section 5.3 highlighted in red font). Additionally, to ensure coherence in the content of the article, we have added a summary of relevant concepts regarding plant phenotypic traits in section 6, enhancing the overall logical coherence of the article.

Once again, we express our gratitude to all the expert reviewers for affirming our work. We look forward to hearing from you again!

Wishing you smooth work and a happy life!

This manuscript is a resubmission of an earlier submission. The following is a list of the peer review reports and author responses from that submission.

Round 1

Reviewer 1 Report

Comments and Suggestions for Authors

The authors propose a review paper about technologies, applications and prospects of high throughput plant phenotyping. The paper perfectly suits the journal scope. Even if the paper could be sufficient to be considered for publication in this journal, I believe it will benefit from major revisions as reported in the following paragraphs.

Starting from the abstract, the authors claim that they'll evaluate the status of the research both "home and abroad". Since the journal is not a specifically domestic Chinese one I suggest to rephrase using "worldwide research, regardless of the country" if needed. However, I missed a detailed discussion and comment about the main differences in high throughput phenotyping with reference to China and other countries.

The authors use the terms "parameters" and "features" interchangeably in the text, while it should be better to decide the notation and keep it stable throughout the text. When dealing with artificial intelligence models, usually the parameters of the model are one thing, while the features are the specific traits (or parts of them) that could be extracted by the AI model.

What is the need of Figure 1d and why the number of articles is plotted with a sort of 3D cumulative representation? Maybe a bar chart should be more appropriate as any permutation on the X-axis represent the same quantities. For the ease of readiness, maybe the labels on the X-axis can be reported in descending order (Stems-Leaves-Roots-Flowers-Fruits/seeds)

NOTE: Starting from the end of page 3 line numbers are no longer available as the same number is repeated many times. So I'll report page and line number as it appears on the manuscript.

Page 5 Line 16. What is a 3D image? I believe it is clearer if the authors refer to 2D images and 3D data

Page 5 line 17. "More complete and intuitive 3D images" It is not clear what an intuitive image is and what is the comparative term for "more complete".

Page 6 line 21. "parameter features". Parameters or features?

Page 6 line 21. "Image processing specifically includes eight steps: ...". This is a really strong claim, as it is not the truth. Even if a large number of image processing software pipelines usually involve the steps reported by the authors, this is not a general rule. For example, grayscale conversion is performed only for solving specific tasks. So please rephrase the sentence according to the scope of the review.

Page 6 line 21. "leading to the development of specific algorithm models". What is an algorithm model? Please clarify.

Page 6 line 22. "transitioning from initial machine learning to deep learning". Please clarify this sentence because the previous paragraph introduces classical computer vision/image processing approaches that should not be confused with machine learning ones. However, all this paragraph seems to me confusing and for a general audience. I suggest a complete rephrase.

Page 7 line 23. Why do the authors define deep learning as a technology? Please explain.

Page 7 line 24. This seems an introduction sentence

Page 7 line 24. "fire primary formats". What the authors mean by format? A format for 3D data could be ".obj", ".ply" either in the binary or ascii format, to name a few. I suggest to use proper vocabulary to describe technical aspects. 

Page 7 Figure 2 caption. Why is the relative path reported in the caption?

Page 8 line 27 and Page 8 Line 30. I am not sure about the choice of presenting results in section 3 (Key technologies) when the authors decided to write a section 4 (Application). I think it will be more appropriate to move results and comments afterwards.

Page 9 line 32. "parameters". Parameters or features?

Page 9 line 36. "more precise phenotypic information". Precise or accurate? Please clarify and explain.

Page 9 line 37. "Renato et al.". Please check if Renato is actually the surname of the first author.

Page 12 Table 3. The authors cite the well known single stage detector YOLO stopping at its early versions, even if there are many recent works that describe the performance improvement also using YOLOv5 or subsequent versions (e.g. v8).

Some examples include:

- Solimani, Firozeh, et al. "Optimizing tomato plant phenotyping detection: Boosting YOLOv8 architecture to tackle data complexity." Computers and Electronics in Agriculture 218 (2024): 108728.

- Wang, Xinfa, et al. "Lightweight SM-YOLOv5 tomato fruit detection algorithm for plant factory." Sensors 23.6 (2023): 3336.

Page 13 section 5. This should be the discussion section but I really missed a comprehensive discussion about the three problems the authors are pointing out. I suggest to give more room to this section.

Page 13 section 5. sub-paragraph titles are sentences rather than titles. Please summarize and put the sentence in the text if needed.

I missed the conclusions of the paper.

Last but not the least, the citation style is not coherent throughout the manuscript. Moreover, many references are reported in the bibliography but I cannot find them in the text. It seems to me a material error so please check thoroughly before resubmitting the manuscript.

Comments on the Quality of English Language

Please check "Quality of English Language"

Author Response

Thank you to the experts for their valuable suggestions! We have completed the modifications according to the requirements and suggestions, and the revised content is highlighted in the original text (Use yellow shading for replies to reviewer 1 and red font for replies to reviewer 2). During the revision process, the author carefully revised the manuscript by reorganizing their ideas and consulting literature. Secondly, it integrates the content of the article, adds important information, and removes unnecessary content, making it easier for readers to understand. At last, the format of the references cited in the article is modified to make the whole article more standardized.

We will now summarize the detailed modification instructions in the revised manuscript (i.e. corresponding responses to each modification suggestion), as shown below:

Comments 1: Starting from the abstract, the authors claim that they'll evaluate the status of the research both "home and abroad". Since the journal is not a specifically domestic Chinese one I suggest to rephrase using "worldwide research, regardless of the country" if needed. However, I missed a detailed discussion and comment about the main differences in high throughput phenotyping with reference to China and other countries.

Response 1: Thank you for the expert review and suggestions. We realize that this extends beyond domestic research, and have accordingly changed the term "domestic and international" to "global" in the abstract section. The discussion on the main differences between domestic and other countries in high-throughput phenotyping has been added to Section 6: Outlook.

Comments 2: The authors use the terms "parameters" and "features" interchangeably in the text, while it should be better to decide the notation and keep it stable throughout the text. When dealing with artificial intelligence models, usually the parameters of the model are one thing, while the features are the specific traits (or parts of them) that could be extracted by the AI model.

Response 2: Thank you for the expert review and suggestions, which we have taken into account. The term "parameters" throughout the entire document has been replaced with "features," maintaining consistency throughout the text.

Comments 3: What is the need of Figure 1d and why the number of articles is plotted with a sort of 3D cumulative representation? Maybe a bar chart should be more appropriate as any permutation on the X-axis represent the same quantities. For the ease of readiness, maybe the labels on the X-axis can be reported in descending order (Stems-Leaves-Roots-Flowers-Fruits/seeds)

Response 3: Thank you for the expert review and suggestions. Figure 1d aims to illustrate the proportion of application of high-throughput phenotyping research in various aboveground organs of crops, providing a foundation for Section 4: Applications. The original use of a 3D cumulative chart has a similar meaning to a bar chart, and the author's intention was to innovate in visualization. However, upon further consideration, it is now apparent that the 3D cumulative chart is not appropriate, as it appears abrupt and lacks rigor. Therefore, we have modified it to a bar chart.

Comments 4: Page 5 Line 16. What is a 3D image? I believe it is clearer if the authors refer to 2D images and 3D data

Response 4: Thank you very much for the expert review and suggestions. In the manuscript, the term "3D images" refers to images with three-dimensional features. However, after consulting literature and careful consideration, we realize that the current understanding of 3D images is not very thorough, as their definition is complex and diverse, which may not be conducive to readers' understanding. In light of the expert's advice, we will change it to: "Construction and Analysis of 2D Images and 3D Data."

Comments 5: Page 5 line 17. "More complete and intuitive 3D images" It is not clear what an intuitive image is and what is the comparative term for "more complete".

Response 5: Thank you very much for the expert review and suggestions. The phrase "more complete" in the original text aims to indicate that researchers may encounter unclear or incomplete images when obtaining 3D images, leading to significant errors in subsequent data analysis. Currently, laser radar technology effectively addresses these issues, thereby improving data accuracy. We understand that there is a lack of comparative terms here, as visible light imaging has been mentioned earlier in the text as being applicable to both two-dimensional and three-dimensional applications. To maintain coherence and logic in the context, we have added "visible light imaging" as a point of comparison.

Comments 6: Page 6 line 21. "parameter features". Parameters or features?

Response 6: Thank you for the expert review and suggestions. This should be about feature extraction.

Comments 7: Page 6 line 21. "Image processing specifically includes eight steps: ...". This is a really strong claim, as it is not the truth. Even if a large number of image processing software pipelines usually involve the steps reported by the authors, this is not a general rule. For example, grayscale conversion is performed only for solving specific tasks. So please rephrase the sentence according to the scope of the review.

Response 7: Thank you for the expert review and suggestions. We understand that the discussion about image processing steps in this section is not uniform or standardized, and may potentially lead to misunderstandings among readers. Therefore, it should not be included in the article. We have deleted this part of the content accordingly.

Comments 8: Page 6 line 21. "leading to the development of specific algorithm models". What is an algorithm model? Please clarify.

Response 8: Thank you for the expert review and suggestions. It appears there may have been a translation error with the term "algorithm model" in the manuscript. The intended meaning was to explain the development of specific algorithms and models. We understand that there is a correlation between algorithms and models, but they have distinct differences: An algorithm is a precise and complete description of a problem-solving method, consisting of a series of clear instructions, while a model, through subjective consciousness, utilizes entities or virtual representations to output the results of algorithms running on data.

Comments 9: Page 6 line 22. "transitioning from initial machine learning to deep learning". Please clarify this sentence because the previous paragraph introduces classical computer vision/image processing approaches that should not be confused with machine learning ones. However, all this paragraph seems to me confusing and for a general audience. I suggest a complete rephrase.

Response 9: Thank you for the expert review and suggestions. Section 3.2 is intended to introduce the intermediate processes involved in phenotypic information acquisition, which mainly include image processing (modified manuscript line 206) and feature extraction (modified manuscript line 210), hence divided into two paragraphs. To avoid readers encountering similar confusion during the reading process, we have added a relevant transitional sentence between these two paragraphs, as detailed in line 209 of the revised manuscript. Additionally, the phrase "transitioning from initial machine learning to deep learning" is intended to elucidate that initially, machine learning methods were primarily used for crop phenotypic feature extraction, while deep learning methods were not widely employed. However, due to the various advantages of deep learning methods at present, research in this area is gradually shifting from using machine learning methods to using deep learning methods. To ensure logical coherence throughout the article, we have modified this sentence to: "As phenomics advances, the methods for feature extraction have become increasingly diverse, and deep learning approaches stand out at this stage," as detailed in line 213 of the revised manuscript.

Comments 10: Page 7 line 23. Why do the authors define deep learning as a technology? Please explain.

Response 10: Thank you for the expert review and suggestions. Defining deep learning as a technology in the original text was a confusion in the author's definition. Deep learning is a learning method within the field of artificial intelligence, with a wide range of applications. Artificial intelligence itself is a technology that encompasses various methods including deep learning, machine learning, natural language processing, and others. Therefore, it should be amended to "deep learning methods."

Comments 11: Page 7 line 24. This seems an introduction sentence

Response 11: Thank you for the expert review. We have summarized five types of image information obtained through 3D reconstruction by consulting a large amount of literature and references, with the most common being point clouds, voxels, and depth maps. Additionally, we have included relevant literature citations for reference.

Comments 12: Page 7 line 24. "fire primary formats". What the authors mean by format? A format for 3D data could be ".obj", ".ply" either in the binary or ascii format, to name a few. I suggest to use proper vocabulary to describe technical aspects. 

Response 12: Thank you for the expert review. We have replaced the term "format" with "type," thus the sentence now reads: "Currently, in the field of 3D reconstruction, five different types of image data are used: implicit, voxel, mesh, point cloud, and depth map." Please refer to line 237 of the revised manuscript for details.

Comments 13: Page 7 Figure 2 caption. Why is the relative path reported in the caption?

Response 13: We want to provide the source files of the drawings to confirm that they are our original graphics, not images referenced from other literature. The relative paths have been removed.

Comments 14: Page 8 line 27 and Page 8 Line 30. I am not sure about the choice of presenting results in section 3 (Key technologies) when the authors decided to write a section 4 (Application). I think it will be more appropriate to move results and comments afterwards.

Response 14: Thank you for the expert review. After careful consideration, the authors believe that the presentation of the relevant results in Section 3 should not be summarized in Section 4. This is because the content intended to be conveyed in these two sections is not entirely consistent. We choose to present the relevant research results in Section 3 to elaborate and validate the article's ideas with certain data, thereby enhancing the coherence and logical flow of the article, and making it easier for readers to understand.

Comments 15: Page 9 line 32. "parameters". Parameters or features?

Response 15: Thank you for the expert review. In line 32 of page 9 of the manuscript (line 338 in the revised manuscript), the term "parameters" has been replaced with "features."

Comments 16: Page 9 line 36. "more precise phenotypic information". Precise or accurate? Please clarify and explain.

Response 16: Thank you for the expert review, we have taken it into account. The authors have confused the definitions of "precision" and "accuracy," as there is a certain difference between the two: accuracy refers to being correct and strictly adhering to facts, standards, or actual situations, while precision refers to refinement to a finer level. Therefore, it should be changed to "accurate."

Comments 17: Page 9 line 37. "Renato et al.". Please check if Renato is actually the surname of the first author.

Response 17: Thank you for the expert review. Upon verification, it was confirmed that Renato is indeed not the first author's surname of the referenced literature. It has been corrected to Casagrande.

Comments 18: Page 12 Table 3. The authors cite the well known single stage detector YOLO stopping at its early versions, even if there are many recent works that describe the performance improvement also using YOLOv5 or subsequent versions (e.g. v8).

Response 18: Thank you for the expert review, and we greatly appreciate the valuable literature resources provided by the expert. We have made the necessary modifications to the content and references in Table 3 accordingly.

Comments 19: Page 13 section 5. This should be the discussion section but I really missed a comprehensive discussion about the three problems the authors are pointing out. I suggest to give more room to this section.

Response 19: Thank you for the expert review. We have made modifications to the content of Section 5, primarily expanding upon the existing content. Please refer to lines 520-608 in the revised manuscript for details.

Comments 20: Page 13 section 5. sub-paragraph titles are sentences rather than titles. Please summarize and put the sentence in the text if needed.

Response 20: Thank you for the expert review. We have revised the subheadings in Section 5 and relocated the original headings to appropriate positions within the text.

Comments 21: missed the conclusions of the paper.

Response 21: Thank you for the expert review, and we have taken the advice into account. We have added Section 6, Outlook, to the end of the article, where we provided insights into the future of high-throughput phenotyping technology for crops, including the main differences between domestic and other countries in this field. Please refer to line 614 of the revised manuscript for details.

Comments 22: Last but not the least, the citation style is not coherent throughout the manuscript. Moreover, many references are reported in the bibliography but I cannot find them in the text. It seems to me a material error so please check thoroughly before resubmitting the manuscript.

Response 22: We appreciate the recommendations from the expert review. We sincerely apologize for any inconvenience the incorrect reference may have caused you. All of the reference formats cited in the manuscript have been updated and converted to a single format. The locations listed in the article's references were also independently checked.

Looking forward to hearing from you again!

Wishing all of you a successful work and a joyful life once again!

Reviewer 2 Report

Comments and Suggestions for Authors

I have reviewed the Review titled "Crop high throughput phenotype: Technologies, Applications and Prospects" by Shuyuan He et al., which has been submitted to the journal Agriculture. Overall, the review aims to provide an overview of the current state of high-throughput phenotyping technology in crop research, its applications, challenges, and potential solutions, with a focus on its significance for advancing agricultural development and breeding practices.

Authors start (Abstract) indicating that "with the rapid growth of agriculture, an increasing number of plant researchers have identified a large number of genes, but there is a serious lack of research on plant phenotypic traits". Improve or correct this idea, since apparently it is not good to "find genes". Start with a paragraph more appropriate to the review. Furthermore, the title of the review says nothing about crop breeding.

Line 55: “Here, we provide an overview of research on high-throughput phenotyping technologies in crops, with a focus on advancements in crop phenomics”, is redundant. HTP is a phenomic technology. Moreover, the authors should clearly outline the benefits of this literature review. Why was it performed?

Line 64: "Theoretical guidance for crop phenotyping research" is vague. It would be beneficial to specify what kind of theoretical guidance is being provided. Include it in the Introduction section. The phrase "accelerating the efficiency of breeding superior varieties and evolution" could be clearer. It's unclear how efficiency is being accelerated and how it relates to the evolution of superior varieties.

In the manuscript, there are many ways in which 'high-throughput phenotyping' (ex. Line 55, 67, 79, 140, etc.) is written. I suggest unifying it and using a common abbreviation (HTP).

Most of the references are from Asia. To enrich the literature for a diverse audience, also consider abundant information from other regions with different types of environments (e.g., Mediterranean climate).

Deep learning techniques are not exclusive for HTP. There are many applications across different fields. Authors should highlight this (for example, in the conclusions, line 628).

Phenomic prediction models are practically absent. Furthermore, nothing is mentioned regarding integrative prediction with other 'omics' data. There is ample evidence in the literature of the benefits of HTP when coupled with genomic information (for example). The authors should include a substantial chapter to this point.

Author Response

Thank you to the experts for their valuable suggestions! We have completed the modifications according to the requirements and suggestions, and the revised content is highlighted in the original text (Use yellow shading for replies to reviewer 1 and red font for replies to reviewer 2). During the revision process, the author carefully revised the manuscript by reorganizing their ideas and consulting literature. Secondly, it integrates the content of the article, adds important information, and removes unnecessary content, making it easier for readers to understand. At last, the format of the references cited in the article is modified to make the whole article more standardized.Then, considering readers from different regions, we have expanded the scope of literature cited.

We will now summarize the detailed modification instructions in the revised manuscript (i.e. corresponding responses to each modification suggestion), as shown below:

Comments 1: Authors start (Abstract) indicating that "with the rapid growth of agriculture, an increasing number of plant researchers have identified a large number of genes, but there is a serious lack of research on plant phenotypic traits". Improve or correct this idea, since apparently it is not good to "find genes". Start with a paragraph more appropriate to the review. Furthermore, the title of the review says nothing about crop breeding.

Response 1: Thank you for the expert review, and we have made the necessary modifications accordingly. We understand the imprecision in the term "seeking genes," so we have changed it to "plant researchers have identified the functions of numerous genes." Additionally, the mention of "crop breeding" in this context is intended to highlight the relative lack of research on crop phenotypes and the crucial role played by high-throughput phenotyping technologies in addressing this gap.

Comments 2: Line 55: “Here, we provide an overview of research on high-throughput phenotyping technologies in crops, with a focus on advancements in crop phenomics”, is redundant. HTP is a phenomic technology. Moreover, the authors should clearly outline the benefits of this literature review. Why was it performed?

Response 2: Thank you for the expert review. The phrase "Here, we provide an overview of research on high-throughput phenotyping technologies in crops, with a focus on advancements in crop phenomics" in line 55 of the original text is essentially redundant. We have made corrections to it, and additional highlights and key content of the article have been added. Please refer to lines 53-59 of the revised manuscript for details.

Comments 3: Line 64: "Theoretical guidance for crop phenotyping research" is vague. It would be beneficial to specify what kind of theoretical guidance is being provided. Include it in the Introduction section. The phrase "accelerating the efficiency of breeding superior varieties and evolution" could be clearer. It's unclear how efficiency is being accelerated and how it relates to the evolution of superior varieties.

Response 3: Thank you for the expert review. We are well aware of the abundance of vague definitions in the original text, which significantly affects the logical flow of the article and makes it difficult for readers to understand. We have made extensive revisions in lines 59-61 of the revised manuscript, removing or improving ambiguous terms.

Comments 4: In the manuscript, there are many ways in which 'high-throughput phenotyping' (ex. Line 55, 67, 79, 140, etc.) is written. I suggest unifying it and using a common abbreviation (HTP).

Response 4: Thank you for the expert review, and we have taken your advice. We have replaced all instances of "high-throughput phenotyping" with the generic acronym "HTP" throughout the manuscript.

Comments 5: Most of the references are from Asia. To enrich the literature for a diverse audience, also consider abundant information from other regions with different types of environments (e.g., Mediterranean climate).

Response 5: We appreciate the advice provided by the professional review. The literature referenced in this study has been examined. The majority of the literature cited consists of well-written, widely cited pieces. At the same time, in order to accommodate readers from various geographic locations, we have broadened the range of literatures acknowledged.

Comments 6: Deep learning techniques are not exclusive for HTP. There are many applications across different fields. Authors should highlight this (for example, in the conclusions, line 628).

Response 6: Thank you for the expert review. In lines 631-633 of the revised manuscript, we have emphasized the widespread application of deep learning methods.

Comments 7: Phenomic prediction models are practically absent. Furthermore, nothing is mentioned regarding integrative prediction with other 'omics' data. There is ample evidence in the literature of the benefits of HTP when coupled with genomic information (for example). The authors should include a substantial chapter to this point.

Response 7: Thank you for the expert review. We understand that an excellent variety breeding requires the interaction of multiple omics and disciplines. High-throughput phenotyping (HTP) should not be limited to genomics but should also integrate with other omics for maximum benefit. Therefore, we have revised the relevant content in the manuscript to emphasize integration with "other omics." However, the focus of this article is to elucidate the current research status and applications of high-throughput phenotyping technology in crop phenotyping. The primary concern is how to rapidly, non-destructively, and accurately obtain crop phenotype information, which is the focus of phenotyping researchers today. While the rapid development of phenomics is beneficial for in-depth research in other omics, dedicating a separate chapter to comprehensive predictive studies with other omics data may appear somewhat abrupt.

Looking forward to hearing from you again!

Wishing all of you a successful work and a joyful life once again!

Round 2

Reviewer 1 Report

Comments and Suggestions for Authors

The authors clearly addressed all the requested points, therefore I recommend the paper for acceptance in its present form.

Comments on the Quality of English Language

English language seems to be good, but since English is not my first language I suggest a double check of the whole manuscript before publication. 

Author Response

We appreciate the recognition from expert reviewers for our work.

We hope this article can provide a research framework for phenotype researchers. The future development of agriculture will accelerate the study of plant phenotypic traits through continuous optimization of system models, improvement of data accuracy, and effective utilization of data.

Wishing all of you a successful work and a joyful life once again!

Best regards!

Reviewer 2 Report

Comments and Suggestions for Authors

In relation to Comment 7 and Response 7.

While the authors acknowledge the importance of integrating multiple omics data for comprehensive phenomic prediction models, their response does not fully address the reviewer's concern.

The reviewer's comment highlights a significant gap in the manuscript: the absence of discussion on phenomic prediction models and integrative prediction with other 'omics' data. While the authors acknowledge the importance of such integration, their response primarily defends the focus of their article on high-throughput phenotyping (HTP) technology in crop phenotyping.

They could consider incorporating a section or at least expanding discussion within the manuscript to cover phenomic prediction models and integrative approaches with other 'omics' data, even if not dedicating an entire separate chapter to it.

Overall, while the authors' response is polite and respectful, it falls short in fully addressing the reviewer's concerns and suggestions. The authors should consider that the topic of breeding is repeatedly mentioned throughout the review, yet it is inadequately addressed in this manuscript.

Author Response

Thank you for the suggestions from expert reviewers. We apologize for not adequately addressing the concerns and suggestions raised by the experts in the last revision. We appreciate the opportunity for another round of revisions provided by the expert reviewers. The issues you have raised regarding this paper are extremely helpful for us to improve its quality.

The specific revisions have been highlighted in red font in the revised manuscript. We hope these changes will enhance the quality of our manuscript.

After careful consideration, we realized the importance of integrating phenotype prediction models with other omics. Therefore, based on our review of relevant literature, we have supplemented the content in Section 4.4 of the revised manuscript. Given the repeated mention of crop breeding throughout the article, we have titled Section 4.4 as "Integration of High-Throughput Phenotyping and Multi-omics in Crops," focusing on the interaction between phenotype and genotype, research progress in predictive models during crop improvement processes, and the construction and application of multi-omics databases.

Furthermore, in the discussion section (Section 5.3), we emphasize the significance of integrating crop phenotyping with multi-omics and call for interdisciplinary collaboration, encouraging the cultivation of interdisciplinary talents.

We would appreciate your consideration of publishing this manuscript and look forward to your response.

Wishing all of you a successful work and a joyful life once again!

Best regards!
